# Hyperalgesia, Increased Temporal Summation and Impaired Inhibitory Mechanisms in Episodic and Chronic Cluster Headache: An Observational Study

**DOI:** 10.3390/biomedicines12020374

**Published:** 2024-02-06

**Authors:** Gabriele Bertotti, Juan Ignacio Elizagaray-García, Jaime Rodríguez-Vico, Alfonso Gil-Martínez

**Affiliations:** 1School of Physiotherapy, Faculty of Health Sciences, Universidad Francisco de Vitoria, 28223 Madrid, Spain; gabriele.bertotti@ufv.es; 2CranioSPain Research Group, Centro Superior de Estudios Universitarios La Salle, Universidad Autónoma de Madrid, 28023 Madrid, Spain; alfonso.gil@lasallecampus.es; 3Headache Unit, Neurology Department, Hospital Universitario Fundación Jiménez Díaz, 28040 Madrid, Spain; jaime.rvico@fjd.es; 4Unidad de Fisioterapia, Hospital Universitario La Paz-Carlos III, IdiPAZ (Hospital La Paz Institute for Health Research), 28029 Madrid, Spain

**Keywords:** cluster headache, central sensitization, hyperalgesia, temporal summation, conditioned pain modulation

## Abstract

Cluster Headache (CH) is a primary headache that causes severe pain. Some evidence suggests that central mechanisms might be involved. The objective of this study was (1) to compare hyperalgesia signs, temporal summation and conditioned pain modulation among episodic (ECH) and chronic CH (CCH) patients and controls, (2) to compare these factors between sides in the patient groups and (3) to compare the psychophysical variables between the groups. This cross-sectional study included 71 subjects divided into three groups (ECH, CCH and controls). Pressure pain thresholds, temporal summation, conditioned pain modulation and other psychosocial variables were measured. The ANOVA showed differences for all physical outcome measures (*p* < 0.05). Bonferroni post hoc analyses showed differences when comparing the patient groups with the healthy subjects (*p* < 0.05), with large effect sizes (*d* > 0.8). No differences between the patient groups were found for almost all the variables (*p* > 0.05). Significant differences for all the variables were detected when comparing the symptomatic and non-symptomatic sides in both the ECH and CCH groups (*p* < 0.05). The ECH and CCH groups showed mechanical hyperalgesia, increased temporal summation and impaired inhibitory mechanisms compared to the controls. Side-to-side differences were also detected within the patient groups. Patients with CCH had poorer sleep quality and quality of life than the controls.

## 1. Introduction

According to the 3rd edition of the International Criteria of Headache Disorders (ICHD-*III*), Cluster Headache (CH) is the most common type of trigeminal–autonomic headache [1]. It affects 0.1% of the general population [2], causing loss of quality of life and socioeconomic burdens [3,4]. Clinically, it is characterized by excruciating pain in the orbital, supraorbital and temporal areas, and it often coexists with autonomic symptoms that usually occur ipsilaterally to the pain, such as lacrimation, conjunctival injection, rhinorrhoea and hyperhidrosis [5]. These symptoms are commonly associated with a sensation of restlessness during attacks, which occur in a series called cluster periods, separated by remission periods that last 3 months or more [1]. As specified by the *ICHD-III* criteria, the chronic form lacks these remission periods and represents 10–15% of CH prevalence [6].

The pathophysiology of CH is complex and not completely understood. Neuroimaging and neuroendocrinological studies have shown an important role played by the hypothalamus [7,8,9] and the trigeminal nucleus caudalis, which are thought to be responsible for mediating the trigeminal–parasympathetic reflex and the autonomic features of CH, respectively [5,10,11]. In addition, maladaptive changes in the somatosensory system involved in pain processing have also been observed [12]. Central sensitization is defined as an increased responsiveness of nociceptive neurons in the central nervous system to their normal or subthreshold afferent [13]. In the literature, it has been suggested that central sensitization could play a major role in CH pathogenesis [14,15,16]. Quantitative sensory testing (QST) is a psychophysical testing method that investigates the functional status of the subject’s somatosensory system [17]. Although central sensitization cannot be directly measured in humans, the QST battery assesses secondary hyperalgesia, temporal summation and allodynia, which are surrogate measures that can suggest central sensitization [18,19]. Several studies have described widespread mechanical hyperalgesia in CH patients [14,15,20,21]. Furthermore, facilitated temporal summation and alterations in the diffuse noxious inhibitory controls of patients with CH have been observed [22,23]. However, some studies assessing temporal summation in CH found no differences compared to healthy subjects [14,24,25].

Although augmented temporal summation and hyperalgesia might suggest central sensitization, other physical methods assessing both pain facilitation and inhibition in CH patients are warranted. To the best of the authors’ knowledge, no cross-sectional study assessing conditioned pain modulation (CPM) has been performed in this population. The CPM is an emerging and reliable method [26] that assesses the diffuse noxious inhibitory controls, which has been widely used in the past decade to evaluate chronic pain conditions [27,28,29,30,31].

One of the major goals of today’s health care is to apply patient-specific approaches. That is why it is of paramount importance to compare patients with different diagnostic subtypes, which would allow clinicians to be more specific when evaluating and diagnosing in clinical practice. Although patients with CH (chronic or episodic) are well defined in the international classification (*ICHD-III*) [1], they are not so well defined from the point of view of somatosensory examination and, therefore, may present some differences that should be considered. Identifying consistent central sensitization signs such as hyperalgesia, facilitated temporal summation, disfunction of diffuse noxious inhibitory controls and other psychosocial variables may promote a better understanding of CH pathogenesis.

Given all the above-mentioned factors, this cross-sectional study aimed (1) to compare hyperalgesia signs, temporal summation and conditioned pain modulation among episodic CH (ECH) and chronic CH (CCH) patients and controls; (2) to compare hyperalgesia signs, temporal summation, and conditioned pain modulation between the symptomatic and the non-symptomatic side within the patient groups; and (3) to compare cutaneous allodynia, pain catastrophizing, neck disability, sleep quality, level of physical activity and quality of life among the episodic and chronic CH patients and the controls.

## 2. Materials and Methods

This cross-sectional study was conducted in two specialized headache clinics in Fundación Jiménez Díaz University Hospital and in La Paz University Hospital, in Madrid. Patients diagnosed with ECH or CCH were recruited through a consecutive sampling method between July and December 2022. Healthy subjects were recruited through advertisements. This study was conducted according to the Strengthening the Reporting of Observational Studies in Epidemiology (STROBE) statement [32]. Furthermore, it was approved by the Ethics Committee for Clinical Research of the two hospitals in Madrid (PI-5213). Before participants could be enrolled, informed consent was obtained.

### 2.1. Participants

Patients were included when meeting the following criteria: adults (>18 years old) diagnosed with strictly unilateral CH (chronic or episodic) by an experienced headache neurologist, according to the *ICHD-III* [1] and lasting 1 year or more prior to participation, with a minimum of 6 h free of attacks (for in-bout episodic and chronic CH). Patients requiring preventive treatment for CH attacks were included. The exclusion criteria were as follows: concomitant diagnosis of any other headache diagnoses (primary or secondary) (*ICHD-III*) [1], invasive occipital nerve stimulation, hypothalamic deep brain stimulation, temporomandibular dysfunction, history of surgery in the cervical–cranio–mandibular region, drug or analgesic medication abuse, meningitis, peripheral neuropathies, rheumatic or systemic disease, alcohol intake 24 h before physical examination, opioid consumption or peripheral anaesthetic block during the previous 4 weeks, exposure to botulinum toxin A during the previous 4 months, a diagnosed major psychiatric disorder or the inability to complete questionnaires, other syndromes suggesting central sensitization (e.g., whiplash, fibromyalgia, complex regional pain syndrome) or other chronic pain diagnosis. Patients experiencing an acute attack of CH were excluded. The control group consisted of healthy subjects older than 18 years who did not meet any of the exclusion criteria listed above.

### 2.2. Procedures

After completing a sociodemographic questionnaire, the physical examination began. Psychosocial questionnaires were completed during breaks between the parts of the examination procedure, which was necessary to avoid sensitization. Following this same principle and trying to prevent an acute attack, the physical testing progressed from less painful to more painful stimuli. The physical examination was conducted by an experienced evaluator blinded to the patient’s condition and lateralization of symptoms. The order of assessment (left or right side) was previously randomized and a five- and ten-minute resting period was respected before changing the side of the exploration and modality [33,34], respectively. The selection process and phases of the examination are schematically represented in Figure 1.

#### 2.2.1. Pressure Pain Thresholds (PPTs)

An algometer (model Fx. 25 Force Gage, Wagner Instruments, Greenwich, CT, USA) was used to measure PPTs, defined as the minimal amount of pressure that would elicit the first sensation of pain. The algometer had a 1 cm^2^ rubber tip attached to a pressure meter [35]. This tool demonstrated a high reliability to quantify PPTs (ICC = 0.91, IC95%: 0.82–0.97) [36]. Participants were positioned in a supine position and a gradually increasing pressure was applied at a rate of 50 KPa/s [17,37,38]. Participants were instructed to respond as soon as the sensation of pressure became painful. PPTs were measured bilaterally in two trigeminal points in the ophthalmic (V_1_) and maxillary (V_2_) nerves, and in two extratrigeminal points, on the median nerve (C5) and tibialis anterior muscle. The order of assessment was randomized between participants. Three consecutive measurements at each anatomical point were assessed with 30 s intervals between each measurement. The mean of the three measurements was calculated for each point and side. Data were recorded in Kg/cm^2^.

#### 2.2.2. Temporal Summation (Wind-Up)

To assess temporal summation, pinpricks were used on the same anatomical points described in the previous section (128 mN pinprick, when tested trigeminally, and 256 mN pinprick, when tested on extratrigeminal points) [17]. First, the perceived intensity of a single pinprick stimulus was quantified with a Visual Analogic Scale (VAS). Secondly, series of 10 repetitive pinprick stimuli of the same physical intensity with an inter-stimulus time of one second (within an area of one cm^2^) were applied and the perceived pain intensity of the last stimulus was quantified with the VAS [17]. The temporal summation magnitude (TSM) was calculated as the difference between the intensity of pain (VAS) induced by the set of 10 stimuli and the intensity of pain (VAS) induced by the first stimulus. High rates for this measurement indicated that the pain perception experienced by the participant had become stronger after repeated stimulation of C-fibers.

#### 2.2.3. Conditioned Pain Modulation (CPM)

To assess descending pain modulation, the following sequential protocol of CPM was used [37]. Subjects were positioned in a sitting position and a first test stimulus (mean of three PPT measurements) was delivered to both the trigeminal (V1) and extratrigeminal areas (tibialis anterior muscle) of one side of the body. Secondly, a thermal conditioning stimulus was delivered by introducing the contralateral hand into cold water (10–12 °C) for 2 min [33,34,38]. The second test stimulus (PPT) was obtained immediately after [37] the conditioning stimulus. The effect of the CPM was calculated by the difference between the second and the first PPT measurements, whereas the percentage of change was obtained with the following formula: % *of change = (second test PPTs* − *first test PPTs)/first test PPTs* [39]. Positive values of the effect of CPM indicated an inhibitory process after the conditioning stimulus, where higher values represented a stronger inhibition and lower values a weaker inhibition capacity.

#### 2.2.4. Psychosocial Questionnaires

Questionnaires were completed during breaks (10 min) (Figure 1), which were introduced between each part of the examination procedure to avoid sensitization processes. The following variables were assessed: cutaneous allodynia with the Allodynia Symptom Checklist (ASC) [40], pain catastrophizing with the Pain Catastrophizing Scale (PCS) [41], neck pain and disability with the Neck Disability Index (NDI) [42], sleep quality with the Pittsburgh Sleep Quality Index (PSQI) [43], physical activity level with the International Physical Activity Questionnaire (IPAQ) [44] and quality of life with the Short Form 12-item Health Survey (SF-12) [45].

### 2.3. Sample Size Calculation

The sample size was calculated using G*Power 3.1 (University of Düsseldorf, Germany). Data from a previous pilot study based on the F-statistic (ANOVA) were used for the sample size. Accepting an alpha error of 0.05, a statistical power of 80% and considering a difference size of 0.36, a total of 70 participants were required for the study. The study groups are not completely balanced due to the reality of clinical practice and the higher percentage of patients with a chronic diagnosis seeking health care in a specialized unit (in a 4th level hospital).

### 2.4. Data Analysis

The data analysis was carried out using IBM SPSS statistical software (v.25.0; SPSS Inc., Chicago, IL, USA). A Shapiro–Wilk test was performed to test data normal distribution (n < 50). The categorical variables were presented as frequencies and percentages, while quantitative descriptive variables were represented by mean and standard deviation.

#### Comparison between Groups and Sides

Clinical variables (e.g., duration, frequency of attacks) were compared between the patient groups using a *t*-test for independent samples. Student’s *t*-test was also used to compare the symptomatic and non-symptomatic side within the headache groups. The effect size of the mean differences was calculated with Cohen’s *d* (0.21–0.50 small, 0.51–0.80 moderate, >0.81 large effects, respectively) [46]. Psychophysical variables were analyzed using a one-way ANOVA and Bonferroni post hoc test. The eta partial squared value (*η_p_*^2^) was calculated from the ANOVA analysis to assess the effect size (*η_p_*^2^ values between 0.01 and 0.039, 0.06 and 0.11, and >0.14 were considered small, medium and large, respectively) [46,47]. When comparing the symptomatic side among the groups, the dominant side of the control group was considered, whereas the non-dominant side of the control group was used to compare with the non-symptomatic side of the patient groups [15,21,48].

## 3. Results

A total of 87 subjects were recruited, of whom 16 participants were excluded because they did not meet the selection criteria. Finally, a total of 71 subjects (14 ECH, 22 CCH and 35 healthy subjects) were physically assessed (Figure 1). There were no differences between the groups regarding demographic characteristics. Regarding headache characteristics, the patient groups showed differences in the length of time since the last attack, but did not differ in pain intensity, duration and frequency of attacks (Table 1).

The one-way ANOVA showed differences for all physical outcome measures. Bonferroni post hoc analyses showed differences when comparing the patient groups with the healthy subjects, with large effect sizes (*d* > 0.8). No differences between the patient groups were found for any of the variables, except for TSM on the tibialis anterior of the symptomatic side (F = 2.29, *p* = 0.038, *d* = 0.68) (Table 2).

When grouping together the symptomatic and non-symptomatic sides of each group and by categorizing all the PPT variables as trigeminal, upper and lower limbs, the one-way ANOVA showed differences among the groups on trigeminal (F = 34.2, *p* < 0.001, η^2^ = 0.5), upper (F = 16.11, *p* < 0.001, η^2^ = 0.32) and lower limbs (F = 13.57, *p* < 0.001, η^2^ = 0.28). Bonferroni post hoc analyses showed no differences between the patient groups on the trigeminal and extratrigeminal regions.

The *t*-test showed differences between PPTs on the symptomatic and non-symptomatic sides within the patient groups, in both the trigeminal and extratrigeminal (tibialis anterior) areas, with moderate to large effect sizes (*d* > 0.5). No side-to-side difference was shown in the PPTs on the median nerve within the ECH and CCH groups (*t* = −0.81; *d* = −0.21 and *t* = −0.09; *d* = −0.02, respectively). The ECH group did not show side-to-side differences in TSM but presented a stronger impairment of the inhibition mechanisms in the trigeminal region of the symptomatic side. The CCH group showed increased TSM on the trigeminal symptomatic side (V_1_), but the endogenous inhibition did not differ between sides (Table 3 and Table 4).

The one-way ANOVA showed differences in two psychosocial outcome measures (PSQI and PCS-12). Bonferroni post hoc analyses showed differences in these variables comparing the CCH group with the healthy subjects, with large effect sizes (*d* > 0.8). No differences between the patient groups were found for any of the psychosocial variables, except for the Physical Score of SF-12 (F = 13.62, *p* = 0.022, *d* = 0.72) (Table 5).

## 4. Discussion

This observational study included and assessed 71 subjects. These results offer a better understanding of the somatosensory function of patients with CH. An amplification of nociceptive signaling and impairment of inhibitory mechanisms were found in the ECH and CCH patients, which could suggest central sensitization.

### 4.1. Pressure Pain Thresholds

According to Arendt-Nielsen et al. (2018), widespread hyperalgesia is a sign compatible with central sensitization [49]. In this regard, widespread hyperalgesia occurs when decreased PPTs are recorded in different body regions, including distal pain-free areas. To analyze this in our study, a comparative analysis of PPTs on the symptomatic side (patient groups) and on the non-symptomatic side in both trigeminal and extratrigeminal regions was performed among the groups. Similarly, a comparison of PPTs on the symptomatic and non-symptomatic side was also performed within the groups.

Differences were observed among all the groups. Significant decreased PPTs of patients with CCH compared to healthy subjects in all trigeminal and extratrigeminal areas of both sides were observed. When comparing the ECH group with healthy subjects, significant reduced PPTs were detected in the trigeminal and extratrigeminal (tibialis anterior) regions on the symptomatic side. Furthermore, decreased PPTs were found in the trigeminal (V_2_) and extratrigeminal (median nerve) region on the asymptomatic side. Overall, this allows us to confirm the presence of generalized hyperalgesia in both groups of patients.

On the other hand, within-group comparative analysis found decreased PPTs on the symptomatic side compared to the non-symptomatic side in the trigeminal areas in both groups of patients (ECH and CCH). Seven studies have compared PPTs on the symptomatic and asymptomatic sides in patients with ECH, with inconclusive results [15,20,21,25,48,50,51]. Palacios-Ceña et al. (2019 and 2020), Gómez-Mayordomo et al. (2020) and Guerrero-Peral et al. (2020) showed no differences between sides, regardless of the trigeminal or extratrigeminal region [20,21,48,50]. Fernandez-de-las-Peñas et al. (2010) found significantly reduced PPTs in zygapophyseal joints C5-C6 and tibialis anterior muscle, but their results did not show significant differences in the trigeminal and other extratrigeminal areas (median, radial and cubital nerves, and on the mastoid process). Ellrich et al. (2006) found increased PPTs on the symptomatic side in the trigeminal region in ECH patients [25]. However, our study showed decreased PPTs in the trigeminal region of patients with ECH, which is consistent with the study performed by Malo-Urriés et al. (2018) that also found decreased PPTs in the trigeminal region on the symptomatic side [51]. Overall, these inconsistencies might be caused by differences in applying the protocols, such as the medication intake allowed, the number of regions assessed and the time of the day of the assessments (e.g., Ellrich et al. (2006) only measured PPTs on V_1_ [25]).

To the best of the authors’ knowledge, only Ellrich et al. (2006) have previously compared mechanical hyperalgesia between sides in patients with CCH [25]. In this regard, our results are inconsistent with those of this study, given that we found decreased PPTs in the trigeminal and extratrigeminal areas, whereas Ellrich et al. (2006) found no differences in the trigeminal PPTs of patients with CCH. Since no other studies comparing PPT measures between sides are available in patients with CCH, more studies are needed to better clarify this inconsistency. In our study, reduced PPTs in the extratrigeminal areas on the symptomatic side were observed only in patients with CCH. This could point to a more widespread pattern of hyperalgesia in the chronic form, which could not be confirmed by the ANOVA when comparing patient groups. Again, more studies assessing the trigeminal and extratrigeminal regions and performing both comparisons (among groups and between sides) are warranted.

The results of the comparison within and among groups allow us to suggest that hyperalgesia is generalized throughout the bodies of patients with ECH and CCH. Although both ECH and CCH groups showed this pattern compared to the healthy controls, no differences between the patient groups were found. Overall, these results support central sensitization as a major underlying mechanism for the pathogenesis of both ECH and CCH, which is in agreement with other studies that suggest that central sensitization mechanisms play a major role in CH pathogenesis [14,15,20,21,48] and in other primary headaches [52,53,54].

### 4.2. Temporal Summation

Temporal summation represents an increase in the perceived intensity of pain in response to sequential stimuli, which reflects the progressive increase in neuronal firing in the dorsal horn in response to repetitive C-fiber stimulation [55]. Again, according to Arendt-Nielsen et al. (2018), a generalized TSM is a sign that can suggest the presence of central sensitization [49].

Our between-groups comparison results showed higher TSM in patients with ECH compared to the controls in the trigeminal (V_2_) and extratrigeminal (median nerve and tibialis anterior) regions on the symptomatic and non-symptomatic sides. Meanwhile, patients with CCH showed only higher TSM in the trigeminal regions (V_1_ and V_2_) on the symptomatic side, but not in the extratrigeminal region and on the non-symptomatic side. On the other hand, within-group comparative analysis showed only differences in patients with CCH in V_1_.

In contrast with other studies, our study found augmented nociceptive facilitation in CH patients compared to healthy subjects [14,24,25]. Since pain TSM measures the amplification of nociceptive signalling, suggesting the presence of central sensitization [56], our results generally show greater facilitation of trigeminal nociception on the symptomatic side of patients with CCH and more widespread (trigeminal and extratrigeminal regions from both symptomatic and non-symptomatic sides) in patients with ECH. Overall, the among-groups and between-sides comparisons showed contradicting results in relation to the analysis of PPTs and CPM, where the CCH group showed more widespread mechanical hyperalgesia features and endogenous inhibitory system impairment. This inconsistency might be related to a methodological reason, since the procedure could not be repeated five times as suggested by the German Research Network on Neuropathic Pain (DFNS) [17]. This could not be carried out for ethical reasons, since patients with CH suffer from one of the most intense forms of pain known [5]. For this reason, and with the intention to avoid or minimize the possibility of triggering an episode, the procedure was performed only once. Furthermore, given that the recent literature indicates a poor correlation between the analysis of endogenous pain modulation through STM and CPM, we could also interpret them as variables that measure different spectra of the same system [57].

### 4.3. Conditioned Pain Modulation

Conditioned pain modulation (CPM) is an emerging psychophysical test that assesses the pain modulation capabilities of nociceptive information [57]. In this study, we presented both the CPM effect and the percentage of change, which is in agreement with the recommendations published by Kennedy et al. (2016), who encouraged this to facilitate the comparison across different protocols assessing CPM [38].

Differences between the ECH group and the controls in pain inhibition in the trigeminal region on the symptomatic side were observed. Additionally, differences in pain inhibition between the CCH group and the controls in the trigeminal region (symptomatic and non-symptomatic sides) and in the extratrigeminal region (non-symptomatic side) were found. Overall, both the ECH and CCH groups showed impaired inhibition of pain, but with a different distribution. This can support the central sensitization theory, where the impairment of inhibitory mechanisms may play a major role [19,55]. However, no differences were found on the symptomatic side (extratrigeminal region) and on the non-symptomatic side of the ECH group. Despite this heterogeneous distribution, an alteration in the inhibition capacity compared to the healthy subjects was observed, which could suggest central sensitization but which might be more segmentally distributed in the ECH patients and which could be specifically related to the spinal trigeminal nucleus. On the other hand, the CCH group manifested a more diffuse disfunction of the inhibitory system of pain, concerning both the trigeminal and extratrigeminal regions on both the symptomatic and non-symptomatic sides compared to the controls. This might point to central sensitization processes occurring both in the spinal cord and in the brain, where the periaqueductal grey has been recently suggested to play a key role in CPM [58]. However, further research is needed to confirm these hypotheses.

Within-group comparative analysis showed that the ECH group presented a stronger impairment of the inhibition mechanisms in the trigeminal region on the symptomatic side. The endogenous inhibition in patients with CCH did not differ between sides. Overall, these results are in line with the above-mentioned pattern: patients with ECH seem to present central sensitization signs that occur in a more segmental manner, whereas patients with CCH appear to show a more widespread hyperalgesia, facilitation of nociception and impairment of inhibition mechanism. This may suggest that central sensitization is a shared neurophysiologic mechanism in both conditions, but its specific pathogenic mechanisms are different and poorly understood.

Overall, when considering all the QST variables assessed in this study, the comparison between the patient groups (ECH vs. CCH) did not show any significant differences for almost any of the variables. Given that both groups showed signs compatible with central sensitization compared to the controls, this supports the hypothesis that central sensitization occurs independently of the condition stage of CH and, consequently, it is a common feature both in its episodic and chronic forms. This would be a differential feature that could better describe CH compared to migraine or a tension-type headache, where chronic types are especially associated with central sensitization compared to episodic forms [54]. However, the lack of somatosensory statistical differences between the patient groups does not allow for a better identification of the specific mechanisms characterizing ECH compared to CCH, which would have been clinically relevant in order to provide new diagnostic insights, as well as providing mechanism-based treatments [59]. Further studies assessing this topic are needed to better differentiate between their somatosensory profiles.

### 4.4. Psychosocial Variables

As is usual in this type of design, where patient groups are compared to each other and to asymptomatic participants, this article found differences between the CCH group and the controls in several variables. One of these variables is as important as quality of life assessed through physical score of SF-12, which showed differences between the patients and the controls, and between the patient groups. Moreover, the CCH patients presented a poorer quality of sleep than the controls, which is in agreement with other studies assessing this topic in CH [60,61,62] and in the migraine population [63]. This might indicate that the hypothalamus and its connection with the autonomic nervous system may play a role in the chronification of the disease. However, this result is more likely to be a consequence of the ongoing attack periods, whereby patients do not reach a remission phase that would improve, even momentarily, their quality of life and possibly their sleep quality. Moreover, recent studies have highlighted that patients with CH suffer prolonged sleep latency and increased time in bed, which are symptoms consistent with insomnia [64]. This has been recently hypothesized to be caused by alterations in the preoptic area of the anterior–superior subunit of the hypothalamus [65], but further studies are necessary in order to confirm its validity. On the other hand, no statistical differences were found in allodynia, pain catastrophizing and neck disability between the patient groups. Moreover, physical activity was similar between the patient groups and the controls. This makes the comparison of the central pain processing variables more valid and unbiased, since an imbalance in physical activity between groups could influence the results for PPTs, TSM and CPM. Recent theories and paradigms in health sciences require psychosocial variables to be recorded and assessed in order to better understand all dimensions of illness from a biopsychosocial point of view.

### 4.5. Limitations

This study has some limitations. Firstly, the QST measures used in this study are surrogates for central sensitization, which allow us to suggest the possibility of the central mechanisms underlying CH pathogenesis, but they do not provide a direct measure of patients’ neurophysiology. Secondly, the protocol to obtain the temporal summation magnitude was only performed once due to ethical and preventive reasons, whereas the DFNS stated it should be performed five times for each subject [17]. Thirdly, we could not differentiate between ECH patients in remission and in a period of attacks, which could also limit the generalization of our findings. Finally, the higher number of CCH patients than of ECH patients because of the availability of patients attending the two specialized headache units may influence the external validity of these results, so they should be translated to clinical practice with caution.

### 4.6. Future Directions

Research on the CH population is progressively increasing. This allows the consistent reporting of hyperalgesia signs. However, only a few studies have assessed temporal summation and with heterogeneous protocols. To the best of our knowledge, this is the first cross-sectional study assessing CPM in patients with CH, where impaired inhibition mechanisms were found. For all these reasons, more cross-sectional studies assessing temporal summation and CPM with homogenous protocols are needed to better understand this primary headache and, consequently, to allow a better prescription of mechanism-based therapies. In modern health systems, it has been proposed that the most effective approaches are those that implement multimodal techniques. In the case of certain types of headaches [e.g., migraine or Cluster Headache], it has been considered that in addition to conservative approaches, certain surgeries may also be of interest [66]. Therefore, the whole range of therapeutic possibilities should be considered when treating such complex patients.

### 4.7. Clinical Implications

Widespread hyperalgesia, augmented temporal summation and the impaired endogenous inhibition of pain are present in patients with CH. Given that these results provide a better understanding of the pathogenesis and that this could drive to new mechanism-based therapies, they are clinically relevant. Some patterns differentiating ECH and CCH somatosensory profiles where identified, but the lack of consistent results did not allow for a detailed differentiation, which could have been helpful for diagnosis and treatment in clinical practice.

## 5. Conclusions

The episodic and chronic CH patient groups showed mechanical hyperalgesia, increased temporal summation and impaired inhibitory mechanisms compared to the controls.

The episodic CH group presented higher hyperalgesia and inhibition impairments in the trigeminal region of the symptomatic side compared to the non-symptomatic side. The chronic CH group showed increased hyperalgesia and nociceptive facilitation in the trigeminal region of the symptomatic side, and more hyperalgesia in the extratrigeminal region compared to the non-symptomatic side.

Patients with chronic CH had poorer sleep quality and quality of life than the controls. Quality of life was also reduced compared to the episodic CH patients. No differences were found between the patient groups in any other psychosocial variable.

## Figures and Tables

**Figure 1 biomedicines-12-00374-f001:**
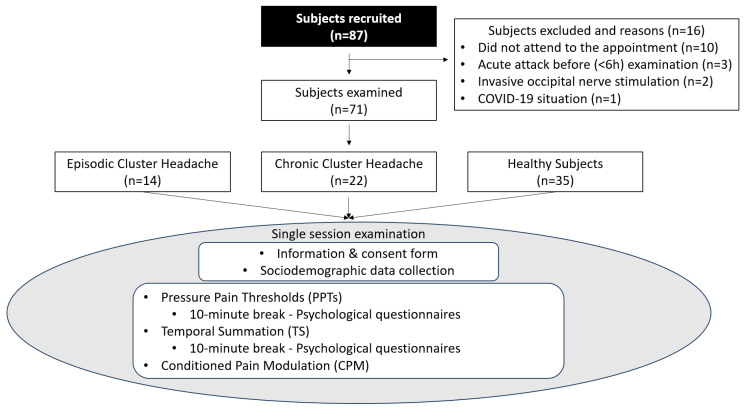
Study flow chart.

**Table 1 biomedicines-12-00374-t001:** Sample characteristics and comparison of headache features between patient groups.

	ECH (n = 14)	CCH (n = 22)	Controls (n = 35)		
	Mean (SD)	Mean (SD)	Mean (SD)	Statistic	*p*-Value
**Age (years)**	50.14 (11.16)	44.05 (11.88)	46.31 (12.64)	F = 1.08	0.34
**BMI (kg/m^2^)**	25.73 (3.16)	25.68 (3.22)	24.96 (2.77)	F = 0.54	0.59
**Time since diagnosis (years)**	13.86 (8.48)	9 (6.27)	-	*t* = 1.97	0.06
**Pain intensity of attacks (VAS)**	9.21 (1.53)	8.98 (1.10)	-	*t* = 0.54	0.59
**Attack number in one day**	3.07 (1.53)	2.40 (1.51)	-	*t* = 1.28	0.21
**Attack duration (minutes)**	49.07 (39.43)	51.59 (39.26)	-	*t* = −0.19	0.85
**Time since last attack (days)**	165 (413) *	9.5 (25) *	-	*t* = 3.66	<0.01
**Pain intensity of last attack (VAS)**	8.93 (2.05)	8.29 (2.06)	-	*t* = 0.9	0.38
	**N (%)**	**N (%)**	**N (%)**		
**Sex (males)**	12 (85.7%)	21 (95.4%)	32 (91.4%)	X^2^ = 1.05	0.59
**Side of attack (right)**	8 (57.1%)	11 (50.0%)	-	X^2^ = 0.17	0.68
**Physical activity (% of sedentary) ^$^**	7 (20.0%)	5 (22.7%)	3 (21.4%)	X^2^ = 0.35	0.98

ECH: Episodic Cluster Headache; CCH: Chronic Cluster Headache; SD: Standard Deviation; BMI: Body Mass Index; VAS: Visual Analogic Scale. * Median (interquartile range). ^$^ Percentage of patients classified as sedentary accorrding to International Physical Activity Questionnaire (IPAQ).

**Table 2 biomedicines-12-00374-t002:** Comparison among groups of physical outcomes.

	ECH (n = 14)	CCH (n = 22)	Controls (n = 35)		Controls vs. ECH	Controls vs. CCH	ECH vs. CCH
	Mean (SD)	Mean (SD)	Mean (SD)	Statistic (Effect Size)	Post hoc Analysis—Cohen’s *d* Effect Size		
**PPT (Symptomatic Side) ^$^**							
**V_1_ *(Ophthalmic Nerve)***	1.00 (0.19)	0.93 (0.25)	1.62 (0.36)	F = 44.06 ** (η_p_^2^ = 0.56)	1.92 **	2.14 **	0.31
**V_2_ *(Maxillary Nerve)***	1.23 (0.22)	1.21 (0.34)	2.03 (0.44)	F† = 42.09 ** (η_p_^2^ = 0.55)	2.04 **	2.03 **	0.07
**MEDIAN NERVE**	6.82 (2.43)	5.36 (2.17)	8.08 (1.18)	F† = 14.96 ** (η_p_^2^ = 0.31)	0.78	1.67 **	0.64
**TIBIALIS ANTERIOR**	6.55 (2.08)	5.76 (1.91)	8.56 (1.64)	F = 17.59 ** (η_p_^2^ = 0.34)	1.13 *	1.60 **	0.40
**PPT (Non-Symptomatic Side) ^#^**							
**V_1_ *(Ophthalmic Nerve)***	1.43 (0.33)	1.24 (0.26)	1.62 (0.32)	F = 10.57 ** (η_p_^2^ = 0.24)	0.59	1.27 **	0.66
**V_2_ *(Maxillary Nerve)***	1.56 (0.28)	1.45 (0.40)	2.04 (0.44)	F = 16.39 ** (η_p_^2^ = 0.32)	1.19 **	1.39 **	0.31
**MEDIAN NERVE**	6.65 (2.00)	5.66 (1.80)	8.05 (1.57)	F = 13.33 ** (η_p_^2^ = 0.28)	0.82 *	1.44 **	0.53
**TIBIALIS ANTERIOR**	7.28 (1.45)	6.35 (2.24)	8.21 (1.40)	F† = 6.76 * (η_p_^2^ = 0.19)	0.66	1.05 **	0.47
**TSM (Symptomatic Side) ^$^**							
**V_1_ *(Ophthalmic Nerve)***	2.21 (1.86)	2.44 (1.25)	1.52 (0.91)	F† = 4.74 * (η_p_^2^ = 0.11)	0.55	0.87 *	0.15
**V_2_ *(Maxillary Nerve)***	2.97 (1.99)	2.82 (1.61)	1.81 (1.01)	F† = 4.93 * (η_p_^2^ = 0.13)	0.86 *	0.79 *	0.08
**MEDIAN NERVE**	2.06 (1.51)	1.68 (1.31)	1.15 (0.71)	F† = 3.36 * (η_p_^2^ = 0.10)	0.91 *	0.54	0.27
**TIBIALIS ANTERIOR**	1.49 (1.92)	0.63 (0.54)	0.46 (0.59)	F† = 2.29 * (η_p_^2^ = 0.14)	0.91 *	0.30	0.68 *
**TSM (Non-Symptomatic Side) ^#^**							
**V_1_ *(Ophthalmic Nerve)***	2.24 (1.68)	1.94 (1.25)	1.29 (0.77)	F† = 3.88 * (η_p_^2^ = 0.11)	0.86 *	0.66	0.21
**V_2_ *(Maxillary Nerve)***	3.06 (2.28)	2.48 (1.82)	1.64 (0.98)	F† = 3.96 * (η_p_^2^ = 0.12)	0.97 *	0.62	0.29
**MEDIAN NERVE**	1.84 (1.38)	1.54 (1.24)	0.95 (0.76)	F† = 3.94 * (η_p_^2^ = 0.11)	0.92 *	0.61	0.23
**TIBIALIS ANTERIOR**	0.86 (1.13)	0.60 (0.64)	0.55 (0.62)	F† = 0.47 (η_p_^2^ = 0.02)	0.39	0.08	0.30
**CPM Effect (Symptomatic Side) ^$^**							
**V_1_ *(Ophthalmic Nerve)***	0.04 (0.10)	0.06 (0.08)	0.25 (0.13)	F† = 26.76 ** (η_p_^2^ = 0.44)	1.64 **	1.67 **	0.15
**TIBIALIS ANTERIOR**	0.35 (1.18)	0.43 (0.67)	0.74 (0.69)	F = 1.65 (η_p_^2^ = 0.05)	0.46	0.46	0.08
**CPM Effect (Non-Symptomatic Side) ^#^**							
**V_1_ *(Ophthalmic Nerve)***	0.23 (0.13)	0.10 (0.11)	0.24 (0.18)	F† = 7.42 * (η_p_^2^ = 0.14)	0.05	0.86 *	1.03
**TIBIALIS ANTERIOR**	0.84 (1.5)	0.29 (0.51)	0.98 (0.94)	F† = 6.56 * (η_p_^2^ = 0.09)	0.12	0.85 *	0.54
**CPM % of change (Symptomatic Side) ^$^**							
**V_1_ *(Ophthalmic Nerve)***	0.05 (0.10)	0.07 (0.10)	0.15 (0.09)	F = 8.59 ** (η_p_^2^ = 0.20)	1.16 *	0.87 *	0.25
**TIBIALIS ANTERIOR**	0.08 (0.22)	0.09 (0.14)	0.08 (0.08)	F† = 0.04 (η_p_^2^ < 0.01)	0.02	0.08	0.07
**CPM % of change (Non-Symptomatic Side) ^#^**							
**V_1_ *(Ophthalmic Nerve)***	0.16 (0.09)	0.09 (0.09)	0.14 (0.09)	F = 2.80 (η_p_^2^ = 0.08)	0.23	0.50	0.77
**TIBIALIS ANTERIOR**	0.14 (0.21)	0.07 (0.09)	0.11 (0.10)	F† = 1.58 (η_p_^2^ = 0.04)	0.14	0.44	0.44

F†: F obtained from statistical Welch’s test; ECH: Episodic Cluster Headache; CCH: Chronic Cluster Headache; SD: Standard Deviation; ^$^: Dominant side for controls; **^#^**: Non-Dominant side for controls; V_1_: Ophthalmic Nerve; V_2_: Maxillary Nerve; * *p* < 0.05; ** *p* < 0.001.

**Table 3 biomedicines-12-00374-t003:** Symptomatic vs. non-symptomatic side comparison within episodic CH group.

		ECH (n = 14) (S) Side	ECH (n = 14) (NS) Side	Statistic	
		Mean (SD)	Mean (SD)	*t* Student	Effect Size
**PPT**	V_1_ (*Ophthalmic Nerve*)	1 (0.19)	1.43 (0.33)	5.84 **	*d* = 1.6
	V_2_ (*Maxillary Nerve*)	1.23 (0.22)	1.57 (0.28)	3.62 *	*d* = 1.35
	MEDIAN NERVE	6.82 (2.43)	6.65 (2.00)	0.39	*d* = 0.08
	TIBIALIS ANTERIOR	6.55 (2.08)	7.28 (1.45)	1.88	*d* = 0.41
**TSM**	V_1_ (*Ophthalmic Nerve*)	2.21 (1.86)	2.24 (1.69)	0.09	*d* = 0.02
	V_2_ (*Maxillary Nerve*)	2.97 (1.99)	3.06 (2.28)	0.3	*d* = 0.04
	MEDIAN NERVE	2.06 (1.51)	1.84 (1.38)	1.11	*d* = 0.15
	TIBIALIS ANTERIOR	1.49 (1.92)	0.86 (1.13)	1.41	*d* = 0.4
**CPM** **EFFECT**	V_1_ (*Ophthalmic Nerve*)	0.04 (0.10)	0.23 (0.13)	5.27 **	*d* = 1.55
	TIBIALIS ANTERIOR	0.354 (1.17)	0.84 (1.49)	1.25	*d* = 0.36
**CPM % OF CHANGE**	V_1_ (*Ophthalmic Nerve*)	0.09 (0.10)	0.16 (0.09)	4.08 **	*d* = 1.23
	TIBIALIS ANTERIOR	0.08 (0.22)	0.14 (0.21)	0.91	*d* = 0.25

ECH: Episodic Cluster Headache; (S): Symptomatic Side; (NS): Non-Symptomatic Side; SD: Standard Deviation; PPT: Pressure Pain Threshold; TSM: Temporal Summation Magnitude; CPM: Conditioned Pain Modulation; V_1_: Ophthalmic Nerve; V_2_: Maxillary Nerve; * *p* < 0.05; ** *p* < 0.001.

**Table 4 biomedicines-12-00374-t004:** Symptomatic vs. non-symptomatic sides comparison within chronic CH group.

		CCH (n = 22) (S) Side	CCH (n = 22) (NS) Side	Statistic	
		Mean (SD)	Mean (SD)	*t* Student	Effect Size
**PPT**	V_1_ (*Ophthalmic Nerve*)	0.93 (0.25)	1.24 (0.26)	8.57 **	*d* = 1.22
	V_2_ (*Maxillary Nerve*)	1.21 (0.34)	1.45(0.4)	3.25 *	*d* = 0.65
	MEDIAN NERVE	5.36 (2.17)	5.66 (1.8)	1.13	*d* = 0.15
	TIBIALIS ANTERIOR	5.76 (1.91)	6.35 (2.24)	2.77 *	*d* = 0.28
**TSM**	V_1_ (*Ophthalmic Nerve*)	2.44 (1.25)	1.94 (1.25)	2.22*	*d* = 0.4
	V_2_ (*Maxillary Nerve*)	2.82 (1.61)	2.48 (1.82)	1.28	*d* = 0.2
	MEDIAN NERVE	1.68 (1.31)	1.54 (1.24)	0.59	*d* = 0.11
	TIBIALIS ANTERIOR	0.63 (0.54)	0.60 (0.64)	0.25	*d* = 0.05
**CPM EFFECT**	V_1_ (*Ophthalmic Nerve*)	0.06 (0.08)	0.10 (0.11)	1.79	*d* = 0.47
	TIBIALIS ANTERIOR	0.43 (0.67)	0.29 (0.51)	0.97	*d* = 0.23
**CPM % OF CHANGE**	V_1_ (*Ophthalmic Nerve*)	0.07 (0.10)	0.09 (0.09)	0.78	*d* = 0.2
	TIBIALIS ANTERIOR	0.09 (0.14)	0.07 (0.10)	0.78	*d* = 0.19

CCH: Chronic Cluster Headache; (S): Symptomatic Side; (NS): Non-Symptomatic Side; SD: Standard Deviation; PPT: Pressure Pain Threshold; TSM: Temporal Summation Magnitude; CPM: Conditioned Pain Modulation; V_1_: Ophthalmic Nerve; V_2_: Maxillary Nerve; * *p* < 0.05; ** *p* < 0.001.

**Table 5 biomedicines-12-00374-t005:** Comparison of psychosocial outcomes among groups.

	ECH (n = 14)	CCH (n = 22)	Controls (n = 35)		Controls vs. ECH	Controls vs. CCH	ECH vs. CCH
	Mean (SD)	Mean (SD)	Mean (SD)	Statistic (Effect Size)	Post hoc Analysis—Cohen’s *d* Effect Size		
**ASC**	4.29 (3.79)	4.55 (3.49)	-	*t* = −0.21 (*d* = 0.07)	-	-	-
**PCS**	35.57 (9.85)	31.32 (8.36)	-	*t* = 1.39 (*d* = 0.47)	-	-	-
**NDI**	6.07 (7.24)	9.64 (7.59)	-	*t* = −1.4 (*d* = 0.48)	-	-	-
**PSQI**	6.86 (3.92)	9.45 (4.94)	4.34 (2.15)	F† = 11.79 ** (η_p_^2^ = 0.29)	0.91	1.46 **	0.57
**PCS-12**	47.94 (9.36)	40.53 (10.86)	52.81 (4.13)	F† = 13.62 ** (η_p_^2^ = 0.33)	0.81	1.65 **	0.72 *
**MCS-12**	52.01 (8.57)	47.10 (11.86)	51.92 (4.02)	F† = 1.66 (η_p_^2^ = 0.07)	0.02	0.6	0.46

F†: F obtained from statistical Welch’s test; ECH: Episodic Cluster Headache; CCH: Chronic Cluster Headache; SD: Standard Deviation; ASC: Allodynia Symptom Checklist; PCS: Pain Catastrophizing Scale; NDI: Neck Disability Index; PSQI: Pittsburgh Sleep Quality Index; PCS-12: Physical Score of SF-12; MCS-12: Mental Score of SF-12; * *p* < 0.05; ** *p* < 0.001.

## Data Availability

Data are unavailable due to privacy restrictions.

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
