# Peer review of "Hyperalgesia, Increased Temporal Summation and Impaired Inhibitory Mechanisms in Episodic and Chronic Cluster Headache: An Observational Study"

_biomedicines, 2024, doi:10.3390/biomedicines12020374_

Round 1

Reviewer 1 Report

Comments and Suggestions for Authors

The Authors performed an interesting cross-sectional study on a sample of 71 individuals, of whom 14 suffered from episodic cluster headache, 22 from the chronic form and the rest were controls. (The abstract incorrectly reports a total of 71 patients, whereas it is 36).

The study was conducted in two University Hospitals in Madrid. The study is sensitive in the analysis of PPTs, temporal summation and CPM. various psycosocial questionnaires were summarised during the breaks of the procedures.

The subject studied is not new, but the study is sound and reveals an interesting aspect of how CCH produces a derangement of sleep rhythms. I would suggest expanding this section and in this regard I list some useful papers: PMID: 38212704 PMID: 37596555 and also a recent overview on CH PMID: 37667192

Reviewer 2 Report

Comments and Suggestions for Authors

The Authors performed an observationsl study on hyperalgesia, increased temporal summation, and impaired inhibitory mechanisms in episodic and chronic cluster headache. The results are somewhat new and interesting to a global audience. The Authors should add, in the "Discussion" section, a brief outline of the role of surgery in temporal localizations. In this regard, the following reference is missing "Raposio G, Raposio E. Temporal surgery for chronic migraine treatment: A minimally-invasive perspective. Ann Med Surg, 2022, 76, 103578.".

Comments on the Quality of English Language

Almost fine
